# Localization of Multiple Jellyfish Toxins Shows Specificity for Functionally Distinct Polyps and Nematocyst Types in a Colonial Hydrozoan

**DOI:** 10.3390/toxins15020149

**Published:** 2023-02-13

**Authors:** Anna M. L. Klompen, Matthew K. Travert, Paulyn Cartwright

**Affiliations:** Department of Ecology and Evolutionary Biology, University of Kansas, Lawrence, KS 66045, USA

**Keywords:** cnidaria, hydrozoa, *Hydractinia*, coloniality, venom, JFTs, nematogenesis, in situ hybridization, immunohistochemistry

## Abstract

*Hydractinia symbiolongicarpus* is a colonial hydrozoan that displays a division of labor through morphologically distinct and functionally specialized polyp types. As with all cnidarians, their venoms are housed in nematocysts, which are scattered across an individual. Here, we investigate the spatial distribution of a specific protein family, jellyfish toxins, in which multiple paralogs are differentially expressed across the functionally specialized polyps. Jellyfish toxins (JFTs) are known pore-forming toxins in the venoms of medically relevant species such as box jellyfish (class Cubozoa), but their role in other medusozoan venoms is less clear. Utilizing a publicly available single-cell dataset, we confirmed that four distinct *H. symbiolongicarpus* JFT paralogs are expressed in nematocyst-associated clusters, supporting these as true venom components in *H. symbiolongicarpus*. In situ hybridization and immunohistochemistry were used to localize the expression of these JFTs across the colony. These expression patterns, in conjunction with known nematocyst type distributions, suggest that two of these JFTs, HsymJFT1c-I and HsymJFT1c-II, are localized to specific types of nematocysts. We further interpret JFT expression patterns in the context of known regions of nematogenesis and differential rates of nematocyst turnover. Overall, we show that JFT expression patterns in *H. symbiolongicarpus* are consistent with the subfunctionalization of JFT paralogs across a partitioned venom system within the colony, such that each JFT is expressed within a specific set of functionally distinct polyp types and, in some cases, specific nematocyst types.

## 1. Introduction

The phylum Cnidaria is characterized by the presence of cnidae or cnidocysts, which are complex intracellular structures housed in specialized cells called cnidocytes (or “stinging cells”) [1,2]. The most typical type of cnidae found in cnidarians are nematocysts (housed within nematocytes). Nematocysts are dispersed across various tissues within an individual but are typically most dominant in the tentacles, where upon contact with predators or prey these structures deploy a complex mixture of toxins known as venom, producing what is commonly known as the jellyfish sting [3,4]. Multiple morphologically distinct types of nematocysts exist, but the general structure includes a capsule that releases a hollow, sometimes barbed, tubule that delivers venom into the target [5]. Most species possess more than one nematocyst type and often multiple types are heterogeneously dispersed across tissues [6,7,8,9,10]. Because nematocyst types appear to vary between functionally distinct tissues, it has generally been assumed that different nematocyst types possess distinct toxin cocktails tailored to specific uses [11,12,13]. If this is the case, then cnidarians may allocate specific toxin repertoires to specific functional tissues by dictating where in the body different nematocyst types are dispersed [13,14]. Thus, the phylum Cnidaria offers a unique opportunity to study how a decentralized venom system partitions toxin expression and deployment across functionally distinct tissues and nematocyst types.

Nematocysts are single-use structures used in prey capture, feeding, and defense, amongst other potential uses, and as such cnidarians are continually producing nematocysts to replace those that have been discharged. Nematogenesis is the process of nematocyst development from a progenitor cell population to mature capsule-containing nematocyte [15,16,17,18]. In hydrozoans (subphylum Medusozoa, class Hydrozoa), nematogenesis typically occurs in a specific region of the body that harbors stem cells (called i-cells in hydrozoans), such as within a distinct region in the body column of *Hydra* [16,19] and within the tentacle bulbs of *Clytia* medusae [20]. Subsequently, developing nematocysts migrate to the region where mature nematocysts are deployed. Previous work indicates that toxin expression is initiated early in nematogenesis and decreases during later developmental stages and maturation [21]. Because nematogenesis is an ongoing process, studying the localization of toxin expression can simultaneously inform the expression dynamics of nematogenesis. Thus, the study of cnidarian venom system partitioning necessitates an integrative approach that includes toxin spatial expression (mRNA and protein), and its relationship with nematocyst type and function within the context of the spatial and temporal properties of nematogenesis.

*Hydractinia symbiolongicarpus* (Buss and Yund, 1989) is a colonial hydrozoan found on gastropod shells inhabited by the hermit crab *Pagurus longicarpus* (Say, 1817), which are distributed on the Northeast coast of the United States. *H. symbiolongicarpus* colonies display a division of labor through morphologically distinct polyp types, including gastrozooids (feeding polyps), gonozooids (reproductive polyps), and dactylozooids (foraging and defensive polyps) (Figure 1). These polyps are interconnected by gastrovascular canals called stolons, which transport nutrients as well as several types of cells (including i-cells and developing nematocytes) throughout the colony. Gastrozooids are the only polyp type to possess a mouth, which is surrounded by long filiform tentacles used to catch and deliver prey to their mouths. Gonozooids and dactylozooids possess short capitate tentacles at their apical end, and while gonozooids are typically found towards the center of the colony, dactylozooids are restricted to the aperture of the gastropod shell and only develop when the shell is inhabited by a hermit crab [22,23]. The specific positioning of dactylozooids on the shell margin is thought to facilitate scavenging leftover food pieces or eggs from the hermit crab to then be passed to surrounding gastrozooids, or potentially defend against invading predators inside the shell [24,25].

Previous work using RNA-sequencing (RNA-Seq) has demonstrated that this division of labor in *H. symbiolongicarpus* is complemented by the differential expression of toxins [26]. One set of differentially expressed toxins are the so-called “jellyfish toxins” (JFTs), which are well known as the dominant class of pore-forming toxins in several species of medically relevant box jellyfish (class Cubozoa) (reviewed in [27,28]), including the infamous *Chironex fleckeri* (Southcott, 1956) [29,30,31]. Klompen et al. [26] found two different JFTs upregulated in the gastrozooid and dactylozooid, respectively, when compared to toxin expression across all three polyp types. These two toxins are HsymJFT1c-I (upregulated in gastrozooids) and HsymJFT1c-II (upregulated in dactylozooids) (Table 1) (see Appendix A). A phylogenetic analysis of JFT genes across medusozoans recovered *HsymJFT1c-I* and *HsymJFT1c-II* within a clade that is sister to a clade of JFTs derived solely from cubozoans, multiple of which display potent hemolytic and cardiotoxic bioactivities [32]. While this phylogenetic placement does not necessarily imply *Hydractinia* JFTs share similar potency, this well supported relationship does suggest that these putative pore-forming toxins are true venom proteins, and that the recruitment of JFTs in cnidarian venoms occurred before the split of Hydrozoa from the rest of the medusozoans [32]. Two other *H. symbiolongicarpus* JFTs were identified in Klompen et al. [26,32], here called *HsymJFT2a* and *HsymJFT2* (Table 1; Appendix A). Both toxin sequences fall within clades of putative JFT sequences to which no homolog has been previously characterized; thus, their role as true venom toxins is unclear [32].

Because *HsymJFT1c-I* and *HsymJFT1c-II* mRNA were found to be upregulated in gastrozooids and dactylozooids, we hypothesize that these JFTs are critical venom components for effective predation and foraging/defense, respectively. Of note, these two polyp types display different nematocyst types. Gastrozooid tentacles possess mostly adherent desmonemes, which are not thought to contain toxins, as well as small euryteles found evenly scattered throughout the gastrozooid tentacles and encircling the mouth (hypostome). In contrast, dactylozooids, as well as gonozooids, possess large euryteles concentrated in the capitate tentacles, yet despite this shared nematocyst type, *HsymJFT1c-II* is only significantly upregulated in dactylozooids (Table 1) [26].

Taken together, *H. symbiolongicarpus* provides a tractable model in which to explore how venoms are partitioned within a colonial cnidarian that possesses functional specialization amongst polyp types. Using a combination of localization strategies, we explore the relationship between nematogenesis and the expression of this putative pore-forming toxin family at both the mRNA and protein levels.

## 2. Results

### 2.1. JFT Expression in Single-Cell RNA-Seq Clusters

A single-cell RNA-sequencing (scRNA-Seq) database has previously been generated for *H. symbiolongicarpus* gastrozooid polyps and is publicly available through the *Hydractinia* Genome Portal Project (https://research.nhgri.nih.gov/hydractinia/, accessed on September 2022) [33]. Given that previous transcriptomic studies identified three of the four *Hydractinia* JFT genes as relatively abundant in gastrozooids (Table 1) [26,32], we searched the scRNA-Seq database to determine if JFT expression co-occurs with the expression of a known nematocyst-specific marker. The scRNA-Seq database revealed two discernable clusters expressing the nematocyst-specific marker *Ncol-1* (for ease of communication, these will be referred to as Cluster 1 and Cluster 2) (Appendix A). All three gastrozooid-associated JFTs were expressed within these *Ncol-1* positive clusters, and only *HsymJFT2* showed low to moderate levels of expression in additional clusters of unknown cell types. The co-occurrence of JFT genes in nematocyte-associated clusters provides additional evidence that these toxins are true venom genes. The fourth JFT, *HsymJFT1c-II* (HyS0015.224), displayed low abundance in gastrozooids based on previous transcriptomics studies, and as expected we found minimal expression against the gastrozooid scRNA-Seq database. However, as with the other JFTs, *HsymJFT1c-II* expression was consistent with one of the *Ncol-1* clusters (Cluster 2), suggesting that all four putative *Hydractinia* JFTs are venom genes.

While the three gastrozooid-associated JFTs co-occurred within the *Ncol-1* clusters, each gene displayed a unique pattern of expression within these clusters: *HsymJFT1c-I* was predominantly expressed in Cluster 1, *HsymJFT2a* expressed in Cluster 2, and *HsymJFT2* was found in both *Ncol-1* Clusters 1 and 2, though with relatively higher expression in Cluster 1 (Appendix A). These two *Ncol-1* clusters may correspond to the two different nematocyst types found in gastrozooids: adherent desmonemes and penetrant, small euryteles. We hypothesize that Cluster 1, which displayed a relatively high abundance of *HsymJFT1c-I* mRNA, corresponds to small euryteles. This would suggest that Cluster 2, where *HsymJFT2a* expression was restricted, corresponds to desmonemes. However, the highly specific expression of this putative toxin is surprising given that desmonemes are not thought to contain toxins. The confirmation of nematocyst identity of Cluster 2 thus awaits additional data. *HsymJFT2* is observed in both Clusters 1 and 2, and has relatively high expression in gonozooids (compared to other JFTs; Table 1); given that gonozooid tentacles only contain large euryteles, this suggests that *JFTHsym2* is likely present in multiple cell types.

### 2.2. JFT Expression Is Consistent with Regions of Nematogenesis in Gastrozooids and Gonozooids

Using colorimetric whole mount in situ hybridization (here referred to as ISH), we were able to visualize the expression of *Ncol-1*, *HsymJFT1c-I*, and *HsymJFT2* within gastrozooids and gonozooids (Figure 2). We found no signal of *HsymJFT2a* for either gastrozooids or gonozooids, however this is not unexpected given this JFT is the most lowly expressed relative to the other JFTs based on RNA-Seq (Table 1). Since *Ncol-1* is expressed early in capsule development in all nematocyst types, it is useful in visualizing the regions of active nematogenesis. In gastrozooids, we observed a band-like pattern in the body column (Figure 2A,B). We also detected a signal in gonozooids, but this pattern was not restricted to a particular region and unevenly distributed across the body column (Figure 2D), consistent with a previously reported *Ncol-1::mScarlet* transgenic line [26].

Within the gastrozooid, both *HsymJFT1c-I* and *HsymJFT2* were detected in a similar band-like pattern in the body column as with *Ncol-1*, overlapping spatially with a region of active nematogenesis (Figure 2E,I). Higher magnification observations showed a similar crescent-shaped patterning as observed in the *Ncol-1* ISH (Figure 2C,F,J), which would indicate these are developing nematocysts. The expression of *HsymJFT1c-I* showed more subtle speckled patterning compared to *Ncol-1* and *HsymJFT2*, suggesting expression occurs only in a subset of developing nematocysts. In the differential expression analysis, *HsymJFT1c-I* mRNA shows lower levels of relative expression (conditional mean, normalized counts = 359.01) compared to *HsymJFT2a* (conditional mean, normalized counts = 2245.62). *HsymJFT1c-I* mRNA also had lower relative expression in the scRNA-Seq dataset compared to *HsymJFT2*, in addition to being highly restricted to a specific *Ncol-1* cluster (Cluster 1) as opposed to *HsymJFT2* (expression in Clusters 1 and 2).

In the early stages of nematogenesis, when both *Ncol-1* and putatively toxin expression are occurring, the capsule shape of all nematocyst types is similar, so it is difficult to distinguish between desmonemes and small euryteles; thus, the specific nematocyst types where *HsymJFT1c-I* and *HsymJFT2* were detected could not be identified. Within gonozooids, only *HsymJFT2* could be detected in a small number of cells scattered throughout the body column (Figure 2K,L), whereas no signal was detected for *HsymJFT1c-I* (Figure 2G,H), consistent with the relative expression values in gonozooids for both JFTs as shown in Table 1.

### 2.3. HsymJFT1c-I and HsymJFT1c-II Toxins Putatively Restricted to Specific Nematocyst Types

We performed whole mount immunohistochemistry (IHC) localizations on gastrozooids, gonozooids, and dactylozooids using two custom antibodies against HsymJFT1c-I and HsymJFT1c-II-specific peptides. We selected these two JFTs given they were significantly upregulated in gastrozooids and dactylozooids, respectively [26]. The IHC enabled us to identify the spatial patterning of these proteins across polyp types, as compared to mRNA detection using ISH. Although penetration by antibodies in mature nematocysts is precluded by the polymerization and hardening of the capsule in the final stages of nematogenesis [3,34,35], we were interested to see if the protein could be detected at a late enough stage of nematogenesis to predict the nematocyst type based on capsule shape.

HsymJFT1c-I staining showed a similar speckled distribution in a band-like pattern across the body column of gastrozooids, similar to the observed localization in *HsymJFT1c-I* ISH (Figure 3A). Increased magnification in the body column similarly shows that only a subset of developing nematocytes express HsymJFT1c-I, and several of these cells display ellipsoid-shaped capsules similar to the capsules of small euryteles (Figure 3B). A small number of cells were also consistently detected in the proximal region of gonozooids (Figure 3C,D), which is not a region of nematogenesis according to ISH. These cells also display a morphological shape and size consistent with small euryteles, which are not found in the tentacles of this polyp type. Given the proximity of this region with the stolon tissue, we interpret these positively stained cells as developing nematocytes in the process of migrating via the stolon to other gastrozooid polyps and not destined for the gonozooid tentacles (see Section 3 Discussion).

Overall, IHC localization suggests that HsymJFT1c-I overlaps with a region of nematogenesis in the mid-body column of gastrozooids, and toxin expression is spatially restricted in gastrozooids and gonozooids to a subset of developing nematocysts that are morphologically consistent with small euryteles. HsymJFT1c-II protein was localized to the body column of dactylozooid polyps along the internal, concave region of the spiral (Figure 4). No antibody staining was detected in the other polyp types. This is consistent with the differential expression data, which showed significant upregulation of *HsymJFT1c-II* mRNA in dactylozooids and minimal expression in the other two polyps (Table 1). Staining typically occurred in the proximal region of the polyp to about two-thirds along the length of the body column. As with HsymJFT1c-I, several cells display elongated shaped capsules reminiscent of euryteles. Since large euryteles are the only nematocyst type found in this polyp [26], IHC localization suggests that HsymJFT1c-II toxin is restricted to this subset of large euryteles (also see Appendix A). A similar pattern of migrating nematocysts along the concave region of the spiral of the body column was observed in a *Ncol-1::mScarlet* colony with induced dactylozooids (Appendix A).

## 3. Discussion

Although jellyfish toxins (JFTs) are well characterized components in the venoms of several cubozoans (e.g., [27,29,31]), their role as venom toxins in other medusozoan species is less clear. In this work, we explored the expression of JFTs at multiple scales within the colonial hydrozoan *Hydractinia symbiolongicarpus*. Using a publicly available single-cell database, we show that three of the four JFTs previously identified in transcriptomic studies of *H. symbiolongicarpus* are predominately restricted to one or more nematocyst-associated clusters, providing strong evidence that these putative pore-forming toxins are also components in the venom of this hydrozoan. The fourth JFT (*HsymJFT2*) is expressed in all nematocyst-associated clusters as well as non-nematocyst clusters, suggesting that it may play a more general role beyond a venom toxin. The relative expression of all three gastrozooid-associated JFTs in the single-cell database also matched the relative expression found in Klompen et al. [26] using bulk RNA-Seq: *HsymJFT2* mRNA shows the highest overall abundance, followed by *HsymJFT1c-I* and then *HsymJFT2a*. However, additional analyses will be required to determine which clusters in the scRNA dataset are associated with specific nematocyst types (e.g., if Cluster 2 (Appendix A) corresponds to desmonemes). To further explore the expression of JFTs in *Hydractinia*, we characterized the spatial expression of three JFTs expressed in *H. symbiolongicarpus* tissues and validated their presence within developing nematocysts, and further confirmed the differential expression of these JFT genes between functionally distinct polyp types. By using antibody staining in conjunction with previously characterized distributions of nematocyst types, we found that two JFT toxins, HsymJFT1c-I and HsymJFT1c-II, appear to be restricted to specific nematocyst types. Our findings are summarized in Figure 5.

*HsymJFT1c-I* mRNA and HsymJFT1c-I protein were both shown to overlap in a band-like pattern in the mid-body column in gastrozooids, a known area of nematogenesis and in line with previous studies (e.g., [26,36,37]). Both *HsymJFT1c-I* mRNA and HsymJFT1c-I protein appeared to be expressed in a subset of developing nematocysts, and ISH showed that the signal for *HsymJFT1c-I* occurs within a lower number of cells compared to *Ncol-1* and *HsymJFT2* (Figure 2). Previous characterizations of *Hydractinia* nematocyst types found that gastrozooids have two types of nematocysts, desmonemes and small euryteles (e.g., [38,39]). Desmonemes are primarily adhesive, and as such likely contain a reduced set of toxins [18], yet these are also the more dominant type of nematocyst in gastrozooids and likely compose most of the newly developing nematocysts in the gastrozooid body column [26]. Based on the restriction of *HsymJFT1c-I* to specific clusters in the scRNA-Seq database (Cluster 1) and the results from our ISH, as well as the morphology of capsules observed in the HsymJFT1c-I stained gastrozooids (Figure 3), HsymJFT1c-I appears to be specific to small euryteles. This is an important consideration given the feeding strategy of *H. symbiolongicarpus* gastrozooids. The desmonemes in the tentacles physically ensnare fast-moving prey and allow rapid delivery to the mouth, which is encircled by small euryteles [26]. These small euryteles around the mouth may provide an area of highly concentrated envenomation that can be directly targeted on the prey to ensure it is adequately subdued. The tentacles also contain small euryteles, though at lower proportions compared to desmonemes. As such, the venom-delivering small euryteles may compensate for the higher proportion of non-venom-containing desmonemes by maintaining a potent or fast-acting venom to ensure an adequately rapid response from captured prey. This suggests that the putative pore-forming toxin HsymJFT1c-I is a critical toxin component in prey capture.

One of the more striking patterns resulting from this work is the specificity of HsymJFT1c-II to the dactylozooid. Dactylozooids have no mouth and are therefore unable to participate in the ingestion of prey, and like gonozooids their short, capitate tentacles are composed solely of large euryteles. Dactylozooids (also called spiral zooids) are only found in colonies on the shells inhabited by hermit crabs [22,23]. This would suggest that there is an inherent relationship with the hermit crab that ties into the ecological role of dactylozooids. It has been suggested these are defensive polyps that perhaps sting their crab hosts to prevent predation or otherwise protect the shell from additional inhabitants, such as annelids that often cohabitate within the shell [25], or that they play a role in foraging food scraps or eggs from the hermit crab [24]. Given that HsymJFT1c-II expression is restricted to dactylozooids, which only display large euryteles, this would suggest that HsymJFT1c-II is specific to large euryteles. However, gonozooids also display large euryteles but were not observed to express *HsymJFT1c-II*, suggesting this JFT is restricted to large euryteles utilized specifically by the dactylozooids. Further characterization into the role of this toxin would provide additional insight into the function of dactylozooid tissues in *Hydractinia* colonies.

While previous transcriptome-based work using whole polyps found the partitioning of several venom-like genes across different polyp types, including JFTs, these observations do not consider the putative role of nematogenesis on this pattern. This work presented an opportunity to integrate studies of toxin expression and variation in regions of nematogenesis. There appears to be a clear correlation with early nematogenesis and toxin expression, such that regions of active nematogenesis are also regions of increased toxin expression. Within *H. symbiolongicarpus*, the early nematogenesis marker *Ncol-1* is expressed in the mid-region of the body column of gastrozooids, within the stolon, and the endoderm of developing planulae in *H. symbiolongicarpus* (see [36] as NCol1, [37]). We observe similar expression patterns in *Ncol-1::mScarlet* animals, but the mScarlet signal additionally remains high in mature nematocysts within the tentacles of gastrozooids, likely due to the rapid migration of developing/newly developed nematocysts from the body column. However, the mature nematocysts in gonozooid tentacles typically display a lower level of signal compared to gastrozooids, suggesting a lower rate of nematocyst turnover [26]. In general, gonozooids in *Ncol-1::mScarlet* animals express lower levels of mScarlet-positive cells within the body column compared to gastrozooids. This matches the pattern observed in our *Ncol-1* ISH for both polyp types (Figure 2). From these multiple observations, we can assume that while some level of nematogenesis may take place in the body column of gonozooids, it is likely the later stages of nematogenesis. Therefore, toxin expression will also be reduced in gonozooids. This matches with observations from ISH for *HsymJFT2*; if gonozooids have a low rate of nematocyst turnover, we would expect a low number of positive cells in the body column as these nematocysts are migrating up to the tentacles, which matches our observations (Figure 2).

Given the early developmental stages of gonozooid nematocysts likely occurs in a separate tissue, such as the stolon, then it would be expected that toxin expression would also be lower in whole body gonozooid tissues when compared to gastrozooids. The stolon is the gastrovascular tubing that connects *Hydractinia* polyps in a colony, which is known to transfer nutrients, i-cells, nematocyst precursors, and in the case of aggressive hyperplastic stolon, mature nematocysts [39]. In *Ncol-1::mScarlet* animals, the stolon displays a consistently high level of signal (summarized in Figure 5), suggesting new nematocysts are being produced constantly. Furthermore, because of the high level of turnover in gastrozooids, the stolon is likely producing nematocysts to be transported to the gastrozooid tentacles in addition to those being produced in the body column of gastrozooid polyps. This would explain the pattern of HsymJFT1c-I antibody staining in gonozooids; the small number of stained cells in the proximal region of the gonozooid are likely a byproduct of being in close proximity to the stolon tissue, where multiple cells are being transferred across the colony, and likely these cells will be transferred and used by gastrozooids. Our RNA-Seq results support this explanation, given that there was little expression of this *HsymJFT1c-I* mRNA in the gonozooid. Unfortunately, the chitinous skeletal material that surrounds mature stolons precludes the detection of expression using IHC. Altogether, this pattern of nematogenesis (and therefore toxin expression) makes it difficult to disentangle the partitioning of venom toxins between gonozooids compared to gastrozooids (and likely dactylozooids) using solely RNA-Seq. Therefore, localization-based approaches (ISH, IHC, transgenesis, etc.) are required to determine the tissue and cell specificity of candidate toxin genes.

Considering this context, it remains striking that dactylozooids display a significantly upregulated expression of *HsymJFT1c-II* mRNA and the consistent presence of HsymJFT1c-II protein. To better understand the role of nematogenesis and nematocyst turnover in dactylozooids, we induced dactylozooid development in *Ncol-1::mScarlet* transgenic animals. While dactylozooids were present in colonies, we observed a higher mScarlet signal in mature nematocysts of the dactylozooid tentacles compared to the gonozooids, suggesting these nematocysts are more recently matured and dactylozooids, as with gastrozooids, have a higher turnover of nematocysts relative to gonozooids. We also observed mScarlet-positive cells along the inner concave region of the spiral of the polyp, as with the observed pattern of HsymJFT1c-II-specific antibody staining (Appendix A). This high rate of turnover and the restricted expression of HsymJFT1c-II in exclusively dactylozooid nematocysts implies that this toxin displays a highly specialized function within *H. symbiolongicarpus*, even given the relatively higher rate of nematogenesis and nematocyst turnover in gastrozooids. The high specificity of HsymJFT1c-II in the dactylozooid suggests that the subfunctionalization of this JFT paralog plays a significant role in the division of labor in this colonial hydrozoan.

## 4. Conclusions

Because of the metabolic cost required to synthesize and deploy toxins, and the additional cost of continually replacing nematocysts by cnidarians, it is likely advantageous to partition different toxins between specific tissues to be used only in the appropriate ecological context. *Hydractinia* possesses multiple venom genes from a putative pore-forming toxin family, jellyfish toxins, which are homologous to toxins found in the venoms of medically relevant box jellyfish. We have shown these toxin paralogs are true venom components that show distinct expression patterns across functionally specific tissues and nematocyst types within this colonial hydrozoan. We also found that the interpretation of similar studies exploring the differential expression of toxins using RNA-Seq, especially in colonial organisms, should consider variation in regions of nematogenesis as well as rates of nematocyst turnover when interpreting venom partitioning.

## 5. Materials and Methods

### 5.1. Genomic JFT Identification and Evaluation of Single-Cell Expression

The four putative *Hydractinia* JFT sequences from [26,32] were first searched against the *H. symbiolongicarpus* protein models using BLASTp using the *Hydractinia* Genome Project Portal [33]. The corresponding genes with the best match (Table 1) were used on the *Hydractinia* Single-cell Browser, which corresponds to a dataset derived from isolated gastrozooids. In addition, the gene model for *Hydractinia* minicollagen-1 (*Ncol-1*), a known nematocyst-specific protein, was used to identify clusters that corresponded to developing nematocysts. Resulting graphs were downloaded as PDF files and the final figure (Appendix A) created using Inkscape v1.2.1 [40].

### 5.2. Animal Care

*Hydractinia symbiolongicarpus* colonies were maintained at room temperature (22–23 °C) at the University of Kansas on glass microscope slides in artificial seawater (ASW) (Red Sea Coral Pro Salt Mix) at ~28–31 ppt. Parental colonies (previously established genomic strains: male 291–10 and female 295–8) and transgenic lines (e.g., *Ncol-1::mScarlet*, [26]) from this study were either maintained in separate 5-gallon tanks with pumps or in custom 4-cup containers with air pumps. Animals were fed 2–3 times each week with 2–3-day-old brine shrimp nauplii (*Artemia* sp.) and given 50–100% water changes after each feeding.

Shells of the dwarf hermit crab *Pagurus longicarpus* with or without established *H. symbiolongicarpus* colonies were purchased from Marine Biological Laboratories (Woods Hole, MA, USA). Animals were maintained in a 15-gallon tank with ASW in the same conditions as the cultured *H. symbiolongicarpus* colonies. Crabs were provided frozen shrimp or mussels 2–3 times a week, followed by 2–3-day-old brine shrimp for hydroid colonies. Water changes were conducted weekly, varying from 10 to 50% depending on water quality. For the induction of dactylozooids, explants from *Ncol-1::mScarlet* colony were attached to cleaned gastropod shells, allowed to grow for ~7 days, and a hermit crab introduced. Within 1–2 days, dactylozooids would initiate development at the edge of the colony (Appendix A).

### 5.3. Tissue Fixation

Colonies were relaxed using menthol crystals and individual polyps were removed from the colonies using a razor blade. Dactylozooids were removed from hermit crab shells in a similar manner while crabs were manually restrained. Care was taken to remove the majority of stolon tissue. Animals were fixed in 4% paraformaldehyde (PFA) in fresh ASW or 1X PBS, either at room temperature (RT) for 1 h or at 4 °C overnight. Tissue was washed three times with 0.3% TritonX-100/PBS (PBSTr) for 30 min at 4 °C. The tissue was either used immediately or stored in 100% MeOH at −20 °C.

### 5.4. cDNA Synthesis and in situ Hybridization (ISH)

cDNA synthesis, probe synthesis, and ISH protocols were adapted from [41,42,43]. The RNA was extracted from ~50 isolated polyps (gastrozooids + male gonozooids) from 3–4 day starved *Hydractinia* colonies using the RNEasy Mini Kit (Qiagen Cat. 74104) or a modified protocol with a TRIzol Reagent (Invitrogen, Cat. 15596026) isolation step. cDNA synthesis was conducted using Superscript IV First-Strand Synthesis System (Invitrogen, Cat. 18091050) with the default protocol, and cDNA validated with a nanodrop and subsequently stored at -20C. The cDNA was used to amplify 428, 537, and 424 bp sequence fragments of *HsymJFT1c-I*, *HsymJFT2a*, and *HsymJFT2*, respectively (see Appendix A). These fragments were cloned into TOPO pCR4 TA Cloning Kit (Invitrogen, Cat. 450030), and then PCR amplified using standard M13 primers and Q5 High-Fidelity 2X Master Mix (NEB, Cat. M0492). The following products were PCR purified using a QIAquick PCR Purification Kit (Qiagen, Cat. 28104) for synthesizing probes. A 182 bp fragment for minicollagen-1 (*Ncol-1*) [37] was synthesized into a pBluescript-ll-KS(1) plasmid from Genscript using their Gene Synthesis and Express Cloning service. Probe synthesis was carried out in a 10 μL reaction using MegaScript T3 (Invitrogen, Cat. AM1338) and T7 (Invitrogen, Cat. AM1333) kits, modified with the addition of DIG-UTP (Roche, Cat. 11209256910) and quantified using a Nanodrop. In situ hybridization was conducted through a modified protocol from [44]. Final concentration of probe for hybridization was 1 ng/uL; hybridization was allowed to incubate at 50 °C for ~28–30 h, with solution replaced with fresh denatured probe after ~20 h of initial incubation. Immunostaining for DIG-labeled probes was detected using 1:5000 anti-DIG-Fab-AP (Roche, Cat. 11214667001), and signal detection was initiated using NBT/BCIP Stock Solution (Roche, Cat. 11681451001). After appropriate signal development, the reaction was stopped by washing in 100% EtOH, post-fixed in 4% PFA/PBS, and mounted in 9:1 glycerol/PBS. Images were taken using Leica MZ16F with attached Lumenera Infinity 5 camera and Infinity Analyze v7 software and stacks produced using HeliconFocus Lite software v8.2.0 [45].

### 5.5. Antibody Design, Immunohistochemistry, and Imaging

For the gastrozooid-expressed JFT, HsymJFT1c-I, polyclonal antibodies were produced by AbClonal using their Rabbit Polyclonal Antibody Development For Protein service. For the dactylozooid-specific JFT, HsymJFT1c-II, polyclonal antibody was designed, expressed, and purified from GenScript Biotech using their PolyExpress Premium Antigen-Specific Affinity Purified pAb service. Goat anti-rabbit IgG (H+L) Cross-Adsorbed Secondary Antibody Alexa Fluor 594 (Invitrogen, Cat. A-11012) was used as the secondary antibody. Anti-human Histone H1 (Leinco Technologies Inc. Prod. # H126) and Goat anti-mouse IgG (H+L), F(ab%)2 fragment, CF 488A antibody Secondary Alexa (Sigma-Aldrich, Cat. SAB4600388) were used as controls. Immunostaining was carried out as in Sanders et al. [41]. The DAPI staining occurred in 9:1 glycerol/PBS media, which reduces the staining in capsules of maturing and mature nematocysts. Epifluorescence images were taken using a Leica DM5000 B with attached Lumenera Infinity 3s and Infinity Analyze v7 software. Confocal images were taken either using a Leica Laser Scanning Confocal Upright Microscope at the University of Kansas Microscopy and Analytical Imaging Core or LeicaTCS SP5 with LAS AF software. Images were processed using Fiji (ImageJ2) v2.9.0/1.53t [46] and all final figures were designed using Inkscape [40]. An alignment of all studied JFTs in this work shows antigentic regions selected for antibody production in Appendix A [47,48]. A lack of HsymJFT1c-II staining in gastrozooids is shown in Appendix A.

## Figures and Tables

**Figure 1 toxins-15-00149-f001:**
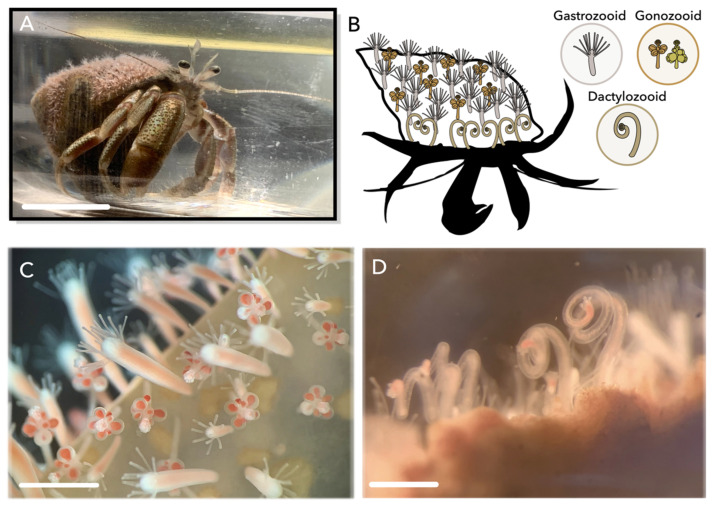
Colonies of *Hydractinia symbiolongicarpus.* (**A**) *Pagurus longicarpus* hermit crab with a pink-tinted *H. symbiolongicarpus* colony growing on the shell. (**B**) Schematic of distribution of polyp types on a colony; gastrozooids scattered throughout the colony, gonozooids found towards the center of the colony, and dactylozooids restricted to the aperture of the hermit crab shell. Stolon tissue is not shown. Hermit crab outline modified from PhyloPic. Icons of each polyp type are shown in right corner. Polyp icons modified from [26]. (**C**) Gastrozooids and gonozooids on hermit crab shell. (**D**) Cluster of dactylozooids at the aperture of a hermit crab shell. Scale bar: 1 cm (**A**), 500 μm (**C**,**D**).

**Figure 2 toxins-15-00149-f002:**
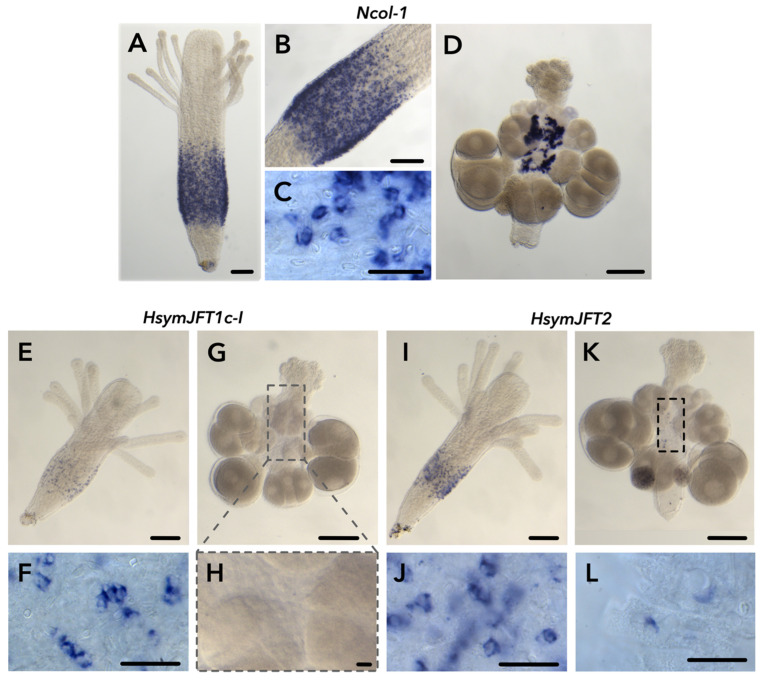
Results of colorimetric in situ hybridization experiments for *Ncol-1*, *HsymJFT1c-I*, and *HsymJFT2*. *Ncol-1* localization is shown for a gastrozooid (**A**), including higher magnification images, (**B**,**C**), and a female gonozooid (**D**). *HsymJFT1c-I* localization is shown as a speckled band-like pattern in the gastrozooid (**E**), high magnification = (**F**), and no signal in the expected region (body column) of gonozooid (**G**). Close-up of the body column of gonozooid, outlined in gray in (**G**,**H**). *HsymJFT2* is localized in a band-like pattern in gastrozooids (**I**), high magnification = (**J**) and in a small number of cells in the body column of gonozooids (**K**), high magnification = (**L**). Outlined box in (**K**) shows region where *HsymJFT2* signal is produced in scattered cells across the gonozooid body column. Scale bars: 100 μm (**A**,**B**,**D**,**E**,**G**,**I**,**K**), and 20 μm (**C**,**F**,**H**,**J**,**L**).

**Figure 3 toxins-15-00149-f003:**
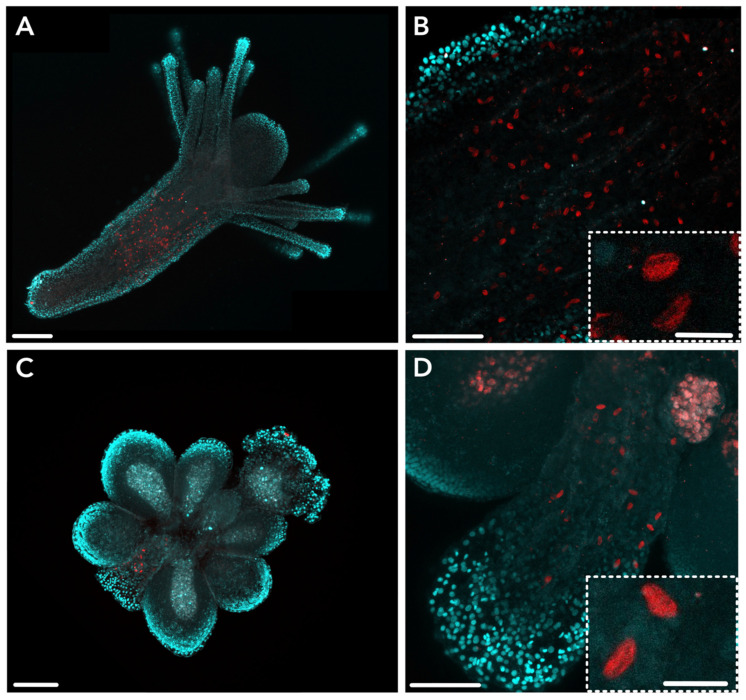
HsymJFT1c-I antibody staining in various tissues of *Hydractinia*. (**A**) HsymJFT1c-I is expressed in the mid-region of gastrozooid polyps. (**B**) Close-up of staining in the gastrozooid body column showing scattered signal. Inset shows higher magnification view of crescent-shaped capsules morphologically similar to euryteles in stained cells. (**C**) HsymJFT1c-I stains a relatively small number of cells in the proximal region of gonozooids. (**D**) Close-up of staining in the gonozooid proximal region shows crescent-shaped capsules. Inset shows higher magnification view of similar crescent-shaped capsules within stained cells as observed in gastrozooid. Red = HsymJFT1c-I, DAPI = cyan. Scale bar: 100 μm (**A**,**C**), 50 μm (**B**,**D**), 10 μm (insets).

**Figure 4 toxins-15-00149-f004:**
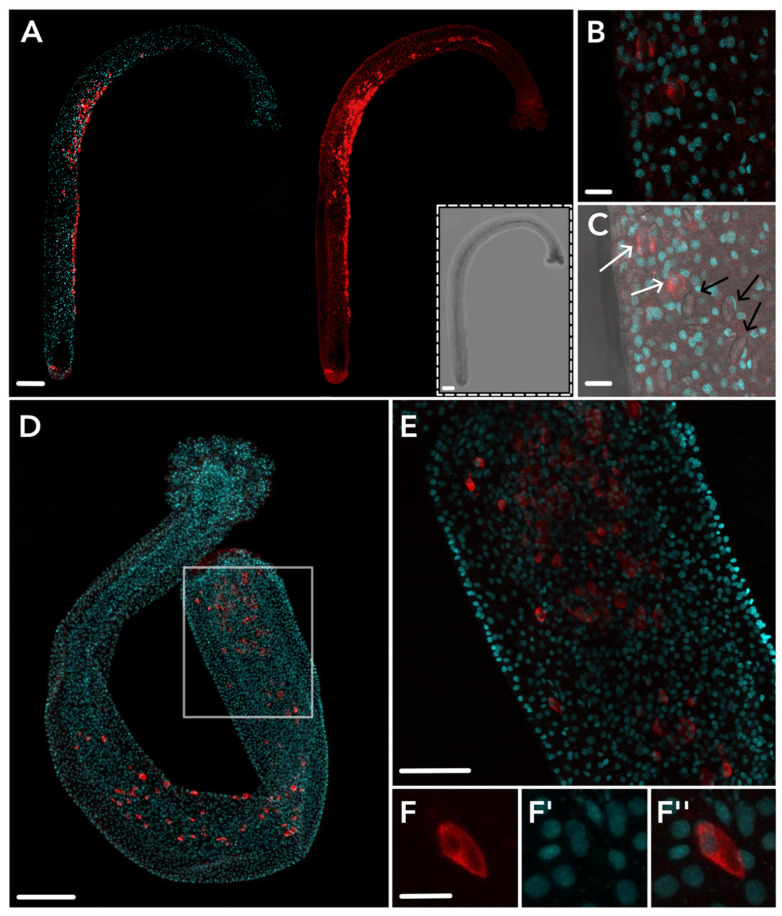
HsymJFT1c-II antibody staining in dactylozooid. Staining across an entire polyp shows consistent staining along the concave spiral of the polyp (**A**); left = DAPI + HsymJFT1c-II, right = HsymJFT1c-II only; inset = bright-field image of polyp. A higher magnification image shows two stained cells with crescent-shaped capsules morphologically similar to large euryteles (**B**). The same image including bright-field shows the same two stained cells in B (white arrows) and three mature large euryteles capsules (black arrows) for comparison (**C**). Proximal staining shown in the white square of a full polyp (**D**) is shown in higher-magnification in (**E**). Panel (**F**–**F″**) shows a single cell within the proximal region with nematocyst capsule forming. No staining was detected in other tissue types. Red = HsymJFT1c-II, DAPI = cyan. Scale bar: 100 μm (**A**,**D**), 50 μm (**E**), and 10 μm (**B**,**C**,**F**).

**Figure 5 toxins-15-00149-f005:**
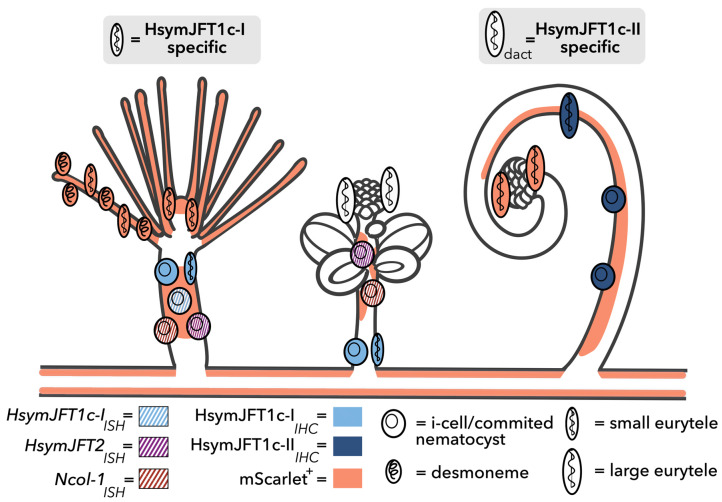
Summary of nematogenesis and JFT localization in *Hydractinia*. Orange overlay shows regions of mScarlet signal based on observations of *Ncol-1::mScar* transgenic. The orange cells in the tentacles and mouth of gastrozooid and tentacles of dactylozooid represent newly matured nematocysts that remain mScarlet-positive (see Appendix A), as opposed to the white cells in the tentacles of the gonozooid that have a reduced signal, corresponding to less recently matured nematocysts. The stolon tissue interconnected to the polyps also displays high mScarlet-positive signal throughout (not shown, but see [26]). The colors (unique for each gene) and pattern (ISH or IHC) for each cell correspond to observations in this work. Gray boxes indicate predictions based on results of this study. Polyp outlines modified from [26].

**Table 1 toxins-15-00149-t001:** Nomenclature and summarization of expression patterns for *Hydractinia* JFTs from previous and current study *.

Toxin Name (This Study)	Gene ID	Cond. Mean, GAST	Cond. Mean, GONO	Cond. Mean, DACT	scRNA-Seq GAST Dataset **	In Situ Hybridization		Protein Localization	
						Polyp	Location	Polyp	Location
HsymJFT1c-I	HyS0055.81	**359.01+**	33.00	68.83	High relative expression; co-localized with *Ncol-1*, cluster 1	GAST	Mid-body column	GAST GONO	Mid-body column Proximal region of the body column
HsymJFT1c-II	HyS0015.224	102.44	26.65	**1486.59+**	Low relative expression; co-localized with *Ncol-1*, cluster 2	-	-	DACT	Body column; concave region of spiral
HsymJFT2a	HyS0028.121	**227.20**	16.03	21.93	Medium relative expression; co-localized with *Ncol-1*, cluster 2	No signal	-	-	
HsymJFT2	HyS0049.112	**2245.62**	522.17	161.37	High relative expression; co-localized with *Ncol-1*, clusters 1 and 2	GAST GONO	Mid-body column Mid-body column	-	-

* Conditional mean values (Cond.Mean) of normalized counts are provided from EBSeq based on n = 4 replicates per polyp type, as shown in [26]. Conditional mean reflects a general expression pattern; values in bold indicate highest relative expression compared to the other two tissue types. + = found to be significantly differentially expressed using EBSeq. GAST = gastrozooid, GONO = gonozooid, DACT = dactylozooid. “-” = did not test/no data. ** Data retrieved from *Hydractinia* Genome Portal Project [33]. Single-cell dataset shown in Appendix A.

## Data Availability

As reported above, a publicly available genome and single-cell database was accessed through the *Hydractinia* Genome Portal Project (https://research.nhgri.nih.gov/hydractinia/, accessed on September 2022). The *Hydractinia* transcriptomic data and JFT phylogeny referenced in this work are publicly available through the listed references.

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
