# Peer review of "Localization of Multiple Jellyfish Toxins Shows Specificity for Functionally Distinct Polyps and Nematocyst Types in a Colonial Hydrozoan"

_toxins, 2023, doi:10.3390/toxins15020149_

Round 1
Reviewer 1 Report
My main concerns are how the figures are presented. The summary Table appears before the actual results.
Figure 2 - I think it would be better suited to try to keep the same flow either across or up and down for the letters. I think putting Like with Like would help in this figure (e.g. A and C) put them next to each other.
Figure 3 - For me the inset image is the most important for these figures, but it's the smallest photo. I think some really nice zoomed in photos of the cells would be great here, the rest doesn't tell you much.
Figure 4 - C is better (compared to those of Figure 3). I think you could in fact merge this figure with Figure 3.
Figure 5 - This figure is too hard to understand. I think the easiest fix would be to use colors to define zones instead of overlapping colors and cell types and patterns. Is NCO1 supposed to be in there? or why is the protein box white? where is this information? Why do we need to know all these things to understand nematogenesis? Nothing in the sexual polyp?
In the text, the authors mentioned that dacylozooids lack a mouth and do not participate in digestion, I guess they meant ingestion rather than digestion.
Reviewer 2 Report
In this study the authors investigated a colonial hydrozoan, Hydractina symbiolongicarpus, that lives on gastropod shells inhabited by the hermit crab Pagurus longicarpus. The investigation was focused on the different spatial distribution of jellyfish toxins across different functionally specialized polyps. By using a public available dataset the authors confirmed the expression of four distinct jellyfish toxins sets of homologous genes which are associated with nematocyst clusters. Applying in situ hybridization and immunohistochemistry allowed the localisation of the jellyfish toxins expressions in the colony and demonstrated that Hydractinia possesses multiple jellyfish toxin proteins that display different distributions in morphologically and functionally different polyp and nematocyst types within the colony.
Overall the article is very well written. The introduction is very clear and well explained with illustrative and relevant figures.
The methodology is well explained and adequate.
The results are very well presented graphically and thoroughly presented.
The discussion is a very deep and well argumented, but sometimes a bit difficult to follow. Perhaps a slight simplification and synthesis might help the reader.
The conclusions followed the presented discussion and are presented clearly.
I would recommend the publication after a very minor simplification of the discussion.
Reviewer 3 Report
The manuscript entitled "Localization of multiple jellyfish toxins shows specificity for functionally distinct polyps and nematocyst types in a colonial hydrozoan" describes the spatial distribution and differential expression of various paralogs of jellyfish toxins (JFTs) across functionally distinct polyps in the colonial hydrozoan Hydractinia symbiolongicarpus. The manuscript is original, well written and I have enjoyed it very much. My congratulations to the authors on a very interesting and well delivered study.
I only have a few comments and suggestions that the authors might want to take into consideration.
- The authors argue in the abstract and discussion that the expression patterns of JFT in different polyp types is consistent with the subfunctionalization of JFT paralogs. Their results show that different paralogs are expressed in polyps that have different ecological functions (i.e., feeding, defense, reproduction), but the toxins themselves might have retained the same ancestral function, that is pore-forming. I am wondering whether the authors think there are specific mutations in the different JFT paralogs that could suggest distinct functions.
- Authors use Ncol-1 as a marker to identify lineages of nematocyst cells in the single cell dataset. I was wondering whether there are additional markers that could be used to identify fully developed nematocysts? It might still not be possible to examine the toxin composition in the developed nematocysts if toxin production happens during nematogenesis, but it would nevertheless be interesting to see whether there are differences in gene expression in the functionally distinct polyps. To examine the venom components of mature nematocysts authors could consider using proteomics, MALDI-IMS and/or immunogold labelling in future studies.
- The scale in Figures 3 and 4 is confusing. It seems that small euryteles in Figure 3 are actually larger than large euryteles in Figure 4? Please clarify this.
- The legend in Figure 5 included Ncol-1 in red, however there is no item colored in red in the figure. I would suggest that authors either remove this, color the corresponding cells in red, or clarify in the legend.
Some minor comments:
- Line 21: change "expression patterns in H. symbiolongicarpus is consistent with" for "are consistent with"
- Line 80: change "(included i-cells and developing nematocytes)" for "(including i-cells and developing nematocytes)"
- Line 91: Table 2 should be Table 1.
- Line 166: change “given gonozoids tentacles only contain” to “given that gonozoid tentacles only contain”
- Line 185: change “alsos” to “also”
- Line 217: change “we were interested if the protein” for “we were interested to see if the protein”
- Line 219: change “and consistent with previous studies” to “in line with previous studies”
- Line 419: it is not very clear what the numbers after the male (291-10) and female (295-8) colonies mean. Please clarify.
Reviewer 4 Report
The authors examine the spatial distribution of several jellyfish toxins (JFTs) in specialized polyp types of the colonial hydrozoan Hydractinia symbiolongicarpus. For their analysis they use single cell datasets, ISH and antibody stainings with polyclonal antibodies targeting HsymJFT1c-I and HsymJFT1c-II. The study provides an interesting case for a functional diversification of toxin paralogs in specific tissues of a colonial cnidarian. The study is well-designed and presented but remains somewhat speculative in the case of dactylozooids. I have listed specific points for improvement below.
Fig. S2: indicate cluster 1 and 2.
Line 185: also
Line 222: nematocytes, not nematocysts
ICH: DAPI usually stains nematocysts by interaction with poly-gamma-glutamate. What is the reason that nematocysts are not stained by DAPI in figures 3 and 4?
Lines 252-255: I was not able to see nematocysts at the concave region of the spiral in the Ncol-1::mScarlet dactylozooids shown in Fig. S3. Please clearly indicate the region in the figure or use better overview figures of single polyps.
Fig. 4: I’m not convinced that capsules are stained here. The shape of the stained region in Fig. 4C and C’’ rather indicates a whole cell. In Fig. 4B you can see one such cell with a distinct cytoneme in the upper left region. To clearly determine that these are nematocytes co-stainings with a capsule marker are necessary.
Also: do the examined toxin sequences contain signal peptides for secretion into the nematocyst vesicle? Please provide an alignment of the amino acid sequences for the toxins examined in the study and indicate signal and possible pro-peptide sequences. Also indicate the epitopes chosen for antibody production and explain the rationale for epitope design.
Lines 292-293: The weaker ISH signal for HsymJFT1c-I does not necessarily indicate that the gene is expressed in a lower number of cells. Other explanations should be considered as well, like a more transient expression or simply technical reasons.
Lines 359-360: is it possible to perform IHC of stolon tissue to clarify this point?
Round 2
Reviewer 1 Report
The changes to the manuscript have greatly improved the readability and presentation of the findings. Overall, I believe this work is a great contribution to the journal and warrants publication. My only lasting criticism is the interpretation of the HsymJFT1c-II data. The single cell data (derived from gastrozoids) suggests this gene is infrequently found in cells of the gastrozooid (and in low abundance). Since the authors suggest this is restricted to the dactylozooids, it would be nice to show a negative result in gastrozooid polyps (Supplement?) to highlight that the Antibody is specific to that protein. I do not doubt that gene is highly up-regulated in dactylozooids, but the derivation of the data (gastrozooid single cell) and the antibody are at odds here.
It would also be nice just to have a negative result for Figure 2 (JFTc-1 data) for the gonozooid data (in close-up photos) just to show completeness, but this is minor.
Finally, Figure 4A is quite difficult to see the two images. Perhaps outlining the body (first option) or putting at transmitted light image nearby (second option) would help us see the structure there.
Overall, the manuscript is a great contribution to our understanding of toxin / cell type interactions in Hydrozoan cnidarians.
Reviewer 4 Report
Most of my concerns were addressed by authors. I have no further comments.
